# Molecular Characterization Analysis and Adaptive Responses of *Spodoptera frugiperda* (Lepidoptera: Noctuidae) to Nutritional and Enzymatic Variabilities in Various Maize Cultivars

**DOI:** 10.3390/plants13050597

**Published:** 2024-02-22

**Authors:** Qiangyan Zhang, Yanlei Zhang, Kexin Zhang, Huiping Liu, Yuping Gou, Chunchun Li, Inzamam Ul Haq, Peter Quandahor, Changzhong Liu

**Affiliations:** 1Biocontrol Engineering Laboratory of Crop Diseases and Pests of Gansu Province, College of Plant Protection, Gansu Agricultural University, Lanzhou 730070, China; zhangqiangyan2@163.com (Q.Z.); 17752269426@163.com (Y.Z.); zhangkx199404@163.com (K.Z.); liuhp110823@163.com (H.L.); gouyp@gsau.edu.cn (Y.G.); 2Dingxi Plant Protection and Quarantine Station, Dingxi 743099, China; 3State Key Laboratory of Ecological Pest Control for Fujian and Taiwan Crops, Fujian Agriculture and Forestry University, Fuzhou 350002, China; 000b370305@fafu.edu.cn; 4CSIR—Savanna Agricultural Research Institute, Tamale P.O. Box 52, Ghana; quandooh@yahoo.com

**Keywords:** *Spodoptera frugiperda*, genotype, special maize, common maize, nutritional indexes, enzyme activities, host adaptation

## Abstract

The fall armyworm, *Spodoptera frugiperda* Smith (Lepidoptera: Noctuidae), a common agricultural pest known for its extensive migration and wide host ranges, causes considerable harm to maize (*Zea mays* L.). In this study, we utilized two molecular marker genes, *COI* and *Tpi*, to compare the genetic characteristics of the collected original samples. Additionally, through an interactive study between *S. frugiperda* larvae and six maize varieties aiming to understand the insect’s adaptability and resistance mechanisms, our analysis revealed that both the *COI* and *Tpi* genes identified *S. frugiperda* as the corn strain. Further examination of the larvae showed significant differences in nutritional indices, digestive, and detoxification enzyme activities. Special maize varieties were found to offer higher efficiency in nutrient conversion and assimilation compared with common varieties. This study revealed adaptations in *S. frugiperda*’s digestive and detoxification processes in response to the different maize varieties. For instance, larvae reared on common maize exhibited elevated amylase and lipase activities. Interestingly, detoxification enzyme activities exhibited different patterns of variation in different maize varieties. The Pearson correlation analysis between nutritional indices, enzyme activities, and the nutritional content and secondary metabolites of maize leaves provided deeper insights into the pest’s adaptability. The results highlighted significant relationships between specific nutritional components in maize and the physiological responses of *S. frugiperda*. Overall, our findings contribute substantially to the understanding of *S. frugiperda*’s host plant adaptability, offering critical insights for the development of sustainable pest management strategies.

## 1. Introduction

The fall armyworm, *Spodoptera frugiperda* Smith (Lepidoptera: Noctuidae), has emerged as a formidable global agricultural menace, originating from the tropical and subtropical zones of the Americas [1,2]. Its remarkable adaptability to various ecological conditions and its ability to cover extensive distances through wind currents have escalated its status to a critical threat to agriculture worldwide. Maize (*Zea mays* L.) is the principal crop worldwide and ranks first among grain crops in terms of production, and pests are the main factors affecting its quality and yield [3]. Special maize generally includes high-lysine maize, glutinous maize, sweet maize, popping maize, high-oil maize, and others. Maize varieties outside of special maize are referred to as common maize [4]. As a polyphagous pest, *S. frugiperda* exhibits a notable preference for maize, with its larval stages causing considerable damage due to its voracious feeding habits across a multitude of host plants [5,6]. The impact on maize is profound, with potential global yield losses estimated between 15% to 73% [7]. In China, the pest was first reported in the southeastern region of Yunnan Province in January 2019. It has since expanded rapidly across the majority of agricultural provinces, inflicting substantial damage to crops, particularly to maize [8]. By the end of the same year, *S. frugiperda* had invaded 26 provinces, afflicting over 1.08 million hectares of crops, thereby presenting a considerable threat to the nation’s maize production [9]. 

Two biotypes of *S. frugiperda*, namely the corn and rice strains, have been identified, exhibiting significant differences in host selection, feeding tendencies, and physiological and behavioral characteristics [10]. The rice type primarily feeds on crops such as rice (*Oryza sativa* L.), alfalfa (*Medicago sativa* L.), and forage grass, while the corn type primarily feeds on maize, sorghum (*Sorghum bicolor* (L.) Moench), and cotton (*Gossypium hirsutum* L.) [11]. Traditional morphological identification poses challenges in distinguishing *S. frugiperda* biotypes, while molecular identification offers a precise and efficient means of achieving accurate classification [12]. Rapidly accurate identification and classification of invasive pests are helpful to advance scientific layout and carry out targeted control measures. Currently, the primary method for identifying *S. frugiperda* biotypes involves the use of molecular marker techniques [13]. The mitochondrial Cytochrome oxidase subunit I gene (*COI*) and Z-chromosome-linked Triose phosphate isomerase gene (*Tpi*) are presently the most commonly used strain markers [14]. Zhang et al. [15] identified 83 samples collected from Yunnan Province, confirming that the invading *S. frugiperda* in China is consistently of the corn type. Upon organically combining these two identification methods and mutually validating them, the identification results can be made more reliable.

The dynamic interaction between plants and herbivorous insects has garnered increasing attention in ecological research because of its complexity and significance [16]. Plants are not passive in the face of herbivory; they can promptly trigger defense responses upon attack, altering the feeding behaviors and oviposition strategies of their assailants. This defensive cascade is balanced by the evolutionary adaptations of phytophagous insects, which employ a repertoire of chemical effectors to neutralize plant chemical defenses through mechanisms such as selective storage, detoxification, and desensitization [17]. The relationship between herbivorous insects and their host plants is shaped by myriad factors, including host species, nutritional conditions, and the insect’s inherent detoxification ability [18,19]. Host plants serve as a source of both nutrition, which varies in quality and quantity [20], and secondary metabolites, which function as a chemical shield against herbivory [21]. Nutritional indices, therefore, become pivotal in evaluating the fitness of herbivores, particularly in species such as *S. frugiperda* [22]. Previous studies have reported that larval consumption and nutrient utilization of host plants can significantly affect the growth, development, and reproduction of *S. frugiperda*, ultimately influencing its host plant adaptability [23,24]. Previous research by Scriber and Slansky Jr [25] highlighted the profound implications of food quality on the physiological processes of insects post-ingestion. Nonetheless, *S. frugiperda* has developed a variety of strategies to cope with the nutritive and defensive properties of host plants, notably through modulating the activities of digestive and detoxification enzymes. These enzymatic adjustments allow the insect to fine-tune its digestive efficiency, broadening its range of potential host plants [26]. Critical enzymes, such as carboxylesterase (CarE), glutathione-S-transferase (GST), and cytochrome P450 (CYP450), are instrumental in the detoxification of plant secondary metabolites, underscoring their importance in the metabolic detoxification processes of insects [27,28]. 

The diversity in nutritional content across various crop species is a critical determinant in the growth, development, and overall survival of insect populations [29]. This is particularly evident in insect larvae, where the quality of the host plant has been shown to significantly differ among plant species, thereby impacting larval health and development [30]. Extensive research into the dietary preferences during the larval stage of *S. frugiperda* has highlighted the significant role of these diets in shaping the pest’s development and reproduction, with maize emerging as a notably suitable host [31,32,33,34]. The influence of different maize varieties and their respective plant tissues on the development and nutritional indices of *S. frugiperda* has been substantiated by numerous studies [35,36,37]. In our previous research, we examined the significant effects of the same maize varieties as in this study on various life history parameters of *S. frugiperda*, such as oviposition preference, developmental duration, pupal weight, survival rate, and fecundity [38]. 

Here, we have identified the collected *S. frugiperda* as the corn strain, and significant variations were observed in their nutritional indices, as well as the digestive and detoxification enzyme activities of *S. frugiperda* on six maize varieties. Furthermore, our research has emphasized the correlation of these parameters with specific nutritional components in maize. These findings elucidate the proficiency of pests in nutrient absorption and their ability to regulate digestion and detoxification processes. The research results are expected to provide valuable insights for effective agricultural practices and pest management, guiding toward more sustainable agricultural approaches.

## 2. Results

### 2.1. Molecular Characterization Analysis of Host Strain of S. frugiperda

Five *COI* (accession no. PP301981-PP301985, GS-1~GS-5) and *Tpi* (accession no. PP331838-PP331842, GS-1~GS-5) gene sequences were obtained from collected *S. frugiperda* in Gansu, and there was no intraspecific sequence variation. Therefore, the first sequence (GS-1) will be chosen for analysis (Figure 1). The *COI* (Figure 1A) and *Tpi* (Figure 1B) gene sequences from GS-1 were closest to the corn strain and 0 bp variation. However, they differed from the rice strain by 17 bp and 10 bp, respectively. Thus, our analysis revealed that both the *COI* and *Tpi* genes identified the collected *S. frugiperda* as the corn strain. 

### 2.2. Effects of Various Maize Varieties on Nutritional Indices of 3rd Instar Larvae of S. frugiperda

Relative Growth Rate (RGR): Our study revealed a significant influence of maize variety on the RGR of *S. frugiperda* larvae (Figure 2A). Larvae feeding on the special maize varieties demonstrated notably higher RGRs compared with those on the common maize varieties (*p* < 0.05), indicating faster growth on the former. Among them, Baitiannuo, Ziyunuo, and Zaocuiwang exhibited the most pronounced RGRs, markedly exceeding those on Wuke 113, Wuke No. 4, and Zhengdan 958 (*p* < 0.05). However, the RGRs on Baitiannuo, Ziyunuo, and Zaocuiwang were statistically indistinguishable from each other (*p* > 0.05). Wuke 113 and Wuke No. 4 had a significantly higher difference in RGR than Zhengdan 958 (*p* < 0.05), but there was no significant difference between Wuke 113 and Wuke No. 4 (*p* > 0.05). The lowest RGR was recorded on Zhengdan 958 (0.395 g/g/d) (Figure 2A). 

Relative Consumption Rate (RCR): *S. frugiperda* displayed the highest RCR on Wuke 113 (7.392 g/g/d), significantly higher than on the other varieties (*p* < 0.05) (Figure 2B). The lowest RCR was observed on Ziyunuo (3.740 g/g/d). However, no statistically significant differences (*p* > 0.05) were found among Wuke No. 4, Baitiannuo, Zaocuiwang, Zhengdan 958, and Ziyunuo (Figure 2B).

Efficiency of Conversion of Digested Food (ECD): ECD values varied significantly among the maize varieties (Figure 2C). Ziyunuo exhibited the highest ECD (28.620%), displaying a significantly greater increase compared with the others (*p* < 0.05). Zaocuiwang ranked second (21.932%), with significantly higher ECD compared with Zhengdan 958, Wuke No. 4, and Wuke 113 (*p* < 0.05). Baitiannuo also showed a significantly higher ECD than Wuke 113. No significant differences (*p* > 0.05) were observed among Baitiannuo, Wuke No. 4, and Zhengdan 958. ECD values for common maize varieties were significantly lower than those for special maize varieties, particularly compared with Zaocuiwang and Ziyunuo (*p* < 0.05). The lowest ECD was on Wuke 113 (8.906%) (Figure 2C). 

Efficiency of Conversion of Ingested Food (ECI): ECI was significantly higher in larvae fed on the special maize varieties compared with common ones (*p* < 0.05) (Figure 2D). Ziyunuo showed the highest ECI (19.395%), significantly surpassing Baitiannuo, Zhengdan 958, Wuke No. 4, and Wuke 113 (*p* < 0.05). Zaocuiwang and Baitiannuo also exhibited significantly higher ECIs compared with Zhengdan 958, Wuke No. 4, and Wuke 113 (*p* < 0.05), with no significant difference between Zaocuiwang and Baitiannuo. The ECIs for common maize varieties were considerably lower, with no significant differences (*p* > 0.05) among Zhengdan 958, Wuke No. 4, and Wuke 113. The lowest ECI was on Wuke 113 (6.670%) (Figure 2D).

Approximate Digestibility (AD): Baitiannuo exhibited the highest AD at 78.559%, whereas Ziyunuo showed the lowest AD at 67.652% (Figure 2E). The AD on Ziyunuo was significantly lower than those on other varieties (*p* < 0.05), but no significant differences (*p* > 0.05) were observed among the cultivars Baitiannuo, Zaocuiwang, Zhengdan 958, Wuke 113, and Wuke No. 4 (Figure 2E).

### 2.3. Effects of Six Maize Varieties on the Digestive Enzyme Activity of S. frugiperda

The digestive enzyme was significantly influenced by maize varieties (Figure 3). The α-amylase activity varied, with a decrease noted from 0.464 U/g in Zaocuiwang to 0.306 U/g in Wuke 113 (Figure 3A). However, this study found no significant differences in α-amylase activity among the larvae reared on the six maize varieties. The pattern of total-amylase activity closely matched that of β-amylase activity across the maize cultivars (Figure 3B,C). The common maize varieties presented a higher β-amylase and total-amylase activity compared with the special maize varieties. Notably, Wuke 113 displayed the highest β-amylase activity (29.370 U/g) and total-amylase (29.677 U/g), which were significantly greater than that in the other varieties (*p* < 0.05). There was a significant increase in β-amylase activity and total-amylase on Wuke No. 4 when compared with Zhengdan 958, Zaocuiwang, Baitiannuo, and Ziyunuo (*p* < 0.05). In contrast, Zaocuiwang exhibited the lowest activity, significantly lower than the others (*p* < 0.05) (Figure 3B,C). The larvae reared on Zhengdan 958 showed the most elevated lipase activity at 19.376 U/g, significantly surpassing those on the other maize varieties (*p* < 0.05) (Figure 3D). Larvae on Wuke 113, Zaocuiwang, Baitiannuo, and Ziyunuo exhibited slightly lower lipase activities, with no significant differences among them. Additionally, the lowest lipase activity was recorded in Wuke No. 4 at 11.695 U/g, significantly less than Zhengdan 958, Wuke 113, and Baitiannuo (*p* < 0.05) (Figure 3D).

### 2.4. Effects of Six Maize Varieties on Detoxification Enzyme Activity of S. frugiperda

Glutathione-S-transferase (GST) Activity: Different maize varieties exerted a significant effect on the GST activity of *S. frugiperda* (*p* < 0.05) (Figure 4A). GST activity notably decreased from 0.205 U/g (Ziyunuo) to 0.112 U/g (Zhengdan 958). The GST activities in Ziyunuo, Wuke 113, and Baitiannuo were significantly higher than those in Zhengdan 958 and Zaocuiwang (*p* < 0.05). Significant variations were observed between Ziyunuo, Wuke 113, and Wuke No. 4 (*p* < 0.05). However, the differences in GST activity among Ziyunuo, Wuke 113, and Baitiannuo, as well as between Baitiannuo and Wuke No. 4, were not significant (*p* > 0.05). Similarly, there were no significant differences between Zhengdan 958, Zaocuiwang, and Wuke No. 4 (*p* > 0.05). 

Cytochrome P450 (CYP450) Activity: Figure 4B reveals the CYP450 activity was significantly affected by the maize cultivars. Wuke 113 (16.895 ng/mL) and Zaocuiwang (17.034 ng/mL) showed the highest activities, significantly surpassing those in Zhengdan 958, Wuke No. 4, Baitiannuo, and Ziyunuo (*p* < 0.05). Both Zhengdan 958 (15.885 ng/mL) and Ziyunuo (15.725 ng/mL) also showed significantly higher CYP450 activity than Wuke No. 4 and Baitiannuo (*p* < 0.05), with Wuke No. 4 showing the lowest activity. The common maize varieties exhibited significant differences in CYP450 activity, with the highest on Wuke 113, followed by Zhengdan 958, and the lowest on Wuke No. 4. For the special maize varieties, Zaocuiwang exhibited the highest activity, followed by Ziyunuo, with the lowest found in Baitiannuo (*p* < 0.05) (Figure 4B). 

Carboxylesterase (CarE) Activity: CarE activity also varied significantly among the maize varieties (Figure 4C). Zhengdan 958 had the highest CarE activity (65.127 U/g), significantly exceeding Wuke No. 4 and Wuke 113 (*p* < 0.05). Zaocuiwang (56.606 U/g) and Baitiannuo (56.425 U/g) exhibited significantly greater CarE activities compared with Wuke No. 4 (*p* < 0.05). There was no significant difference in CarE activity among Zhengdan 958, Zaocuiwang, Baitiannuo, and Ziyunuo or among Wuke No. 4, Wuke 113, and Ziyunuo (*p* > 0.05). Wuke No. 4 had the lowest CarE activity of 38.938 U/g, significantly lower than Zhengdan 958, Zaocuiwang, and Baitiannuo (*p* < 0.05). Among the common maize varieties, the highest activity was on Zhengdan 958, significantly greater than Wuke No. 4 and Wuke 113, with the lowest on Wuke No. 4 (*p* < 0.05). Surprisingly, among the special maize varieties, CarE activity did not differ significantly among Zaocuiwang, Baitiannuo, and Ziyunuo (*p* > 0.05) (Figure 4C).

### 2.5. Correlation among Chemical Substances in Maize Leaves, the Nutritional Indices, Digestive, and Detoxifying Enzymes of S. frugiperda

Pearson’s correlation analysis was employed to discern the relationships between various biochemical parameters in maize leaves, as measured in our previous study (Appendix A) [39], and the corresponding physiological responses of *S. frugiperda* (Figure 5). This analysis yielded several noteworthy correlations:

RGR was extremely significantly correlated with fatty acid (*p* ≤ 0.01, *r* = 0.915), ECD (*p* ≤ 0.01, *r* = 0.727), and ECI (*p* ≤ 0.01, *r* = 0.763). Reducing sugar was also significantly positively linked to RGR (*p* ≤ 0.05, *r* = 0.476). Conversely, RGR had a significantly negative correlation with protein (*p* ≤ 0.05, *r* = −0.559), chlorophyll a (*p* ≤ 0.05, *r* = −0.483), chlorophyll b (*p* ≤ 0.05, *r* = −0.535), β-amylase (*p* ≤ 0.05, *r* = −0.485), and total-amylase (*p* ≤ 0.05, *r* = −0.483), and an extremely significantly negative correlation with amino acids (*p* ≤ 0.01, *r* = −0.715) and moisture (*p* ≤ 0.01, *r* = −0.67), respectively. Similarly, RCR exhibited a significant positive correlation with AD (*p* ≤ 0.05, *r* = 0.512), β-amylase (*p* ≤ 0.05, *r* = 0.515), and total-amylase activities (*p* ≤ 0.05, r = 0.519), and an extremely significant positive correlation with moisture (*p* ≤ 0.01, r = 0.695). However, there were significant negative correlations with fatty acid (*p* ≤ 0.05, *r* = −0.518) and flavone (*p* ≤ 0.05, *p* ≤ 0.05, *r* = −0.487), and a highly significant negative relationship with reducing sugar (*p* ≤ 0.01, *r* = −0.697), ECI (*p* ≤ 0.01, *r* = −0.792), and ECD (*p* ≤ 0.01, *r* = −0.819). ECD showed a highly significant positive correlation with fatty acid (*p* ≤ 0.01, *r* = 0.798), reducing sugar (*p* ≤ 0.01, *r* = 0.666), and ECI (*p* ≤ 0.01, *r* = 0.986), respectively. ECD showed extremely significant negative correlations with amino acids (*p* ≤ 0.01, *r* = −0.607) and moisture (*p* ≤ 0.01, *r* = −0.833) and significant negative correlations with protein (*p* ≤ 0.05, *r* = −0.577), AD (*p* ≤ 0.05, *r* = −0.531), β-amylase (*p* ≤ 0.05, *r* = −0.547), and total-amylase (*p* ≤ 0.05, *r* = −0.55). Extremely significantly positive correlations were present between ECI and fatty acid (*p* ≤ 0.01, *r* = 0.834) and reducing sugar (*p* ≤ 0.01, *r* = 0.618). Conversely, ECI exhibited highly significant negative correlations with amino acids (*p* ≤ 0.01, *r* = −0.6), moisture (*p* ≤ 0.01, *r* = −0.814), β-amylase (*p* ≤ 0.01, *r* = −0.625), and total-amylase (*p* ≤ 0.01, *r* = −0.627), and a significant negative correlation with protein (*p* ≤ 0.05, *r* = −0.553). A significant negative correlation was found between AD and reducing sugar (*p* ≤ 0.05, *r* = −0.583). 

α-amylase and CarE (*p* ≤ 0.05, *r* = 0.51) showed a significant positive correlation. β-amylase and total-amylase activities showed significant positive correlations with moisture and a negative correlation with fatty acid content. Lipase activity was positively correlated with moisture content (*p* ≤ 0.05, *r* = 0.498) and protein levels (*p* ≤ 0.01, *r* = 0.639), the latter being highly significant. 

GST activity had an extremely significant positive correlation with tannin content (*p* ≤ 0.01, *r* = 0.657) but significant negative correlations with amino acids (*p* ≤ 0.05, *r* = −0.578) and chlorophyll b (*p* ≤ 0.05, *r* = −0.53). CYP450 activity had significant positive associations with protein (*p* ≤ 0.05, *r* = 0.539) and tannin (*p* ≤ 0.05, *r* = 0.564) but was significantly negatively correlated with chlorophyll b (*p* ≤ 0.05, *r* = −0.543), and an extremely significant negative correlation with chlorophyll a (*p* ≤ 0.01, *r* = −0.744) and carotenoids (*p* ≤ 0.01, *r* = −0.6). These correlations suggest intricate interactions between the maize plant’s chemical composition and the physiological and metabolic responses of *S. frugiperda*, highlighting the complexity of host–pest interactions.

## 3. Discussion

*S. frugiperda*, due to its wide host range and strong migratory and reproductive ability, has caused serious damage to many crops in China [40]. Genetic studies have identified the invasive *S. frugiperda* populations in China as the corn strain [41]. Guo et al. [23] determined the corn genotype of *S. frugiperda* after 13 generations of feeding on rice or corn. This is consistent with the findings of our study, where alignment analysis based on *COI* and *Tpi* gene fragment sequences revealed complete consistency with the corn type at all sites. Considering that China is the second-largest global maize producer, with cultivation spanning all provinces [42,43], the recurrent incursions of *S. frugiperda* into the country’s maize belts significantly jeopardize its agricultural output and food security. Therefore, it is crucial to understand the physiological changes in *S. frugiperda* after feeding on maize. However, due to limitations in sampling locations and sample numbers, the existence of the rice type of *S. frugiperda* cannot be completely ruled out. Therefore, it is necessary to conduct larger-scale monitoring and investigations.

The nutritional indices play a pivotal role in reflecting the nutrient utilization of insects to host plants [25]. Polyphagous insects, which exhibit a diverse range of host plants, demonstrate significant differences in their adaptability and ability to utilize nutrients [44]. Our study elucidated notable distinctions in the nutritional indices of *S. frugiperda* reared on six maize varieties. These variations were further reflected in the fitness levels observed among the maize varieties. Particularly, the special maize varieties exhibited markedly higher values for ECD, ECI, and RGR compared with the common maize varieties. *S. frugiperda* larvae showed a more efficient conversion of digested food into biomass when feeding on the special maize varieties compared with common ones. Similarly, Hoo and Fraenkel [45] reported that larvae consume smaller quantities of specific food sources due to their increased efficiency in converting it, thereby achieving the desired level of growth without the need for large amounts of food. Interestingly, Wuke 113 displayed the highest RCR yet the lowest ECD and ECI, which is similar to the findings of Behmer [46] and Silva et al. [47]. This shows that the compensatory feeding behavior exhibited by insects when consuming hosts with lower nutritional value, such as prolonged feeding time or increased food intake. More importantly, Pearson correlation analysis revealed that nutrient substance contents and secondary metabolites of maize leaf can significantly affect nutritional indexes. For instance, higher fatty acid and reducing sugar, and lower moisture and protein can lead to higher RGR, ECD, and ECI, yet the lowest RCR of the special maize varieties. The observed phenomenon suggests that the special maize varieties probably contained sufficient nutrients, which helped *S. frugiperda* larvae to adapt and efficiently utilize its nutrients. This was demonstrated by a higher relative growth rate of *S. frugiperda* in comparison with the common maize varieties. Other studies have also demonstrated lower ECI and ECD in *Spodoptera littoralis* Boisd (Lepidoptera: Noctuidae) larvae reared on Mashhad cowpea (*Vigna sinensis* L.) cultivar [48]. Similarly, Kang [49] found that a high proportion of reducing sugars in plants was conducive to feeding larvae and adults, and promoted the growth and metabolism of insects. Moreover, low-protein diets influenced the developmental period, larval weight and mortality, pupation rate, percentage of adult emergence, and nutritional indices [50,51]. In our study, Ziyunuo showed the lowest AD and RCR but the highest ECD and ECI among the tested maize varieties. This discrepancy could be due to the high content of flavonoid metabolites in Ziyunuo, which possibly inhibited the approximate digestibility and consumption of *S. frugiperda* larvae [39]. Similar to *Hyphantria cunea* Drury (Lepidoptera: Arctiidae) larvae, *S. frugiperda* larvae might reduce the toxicity of phenolic secondary metabolites by reducing their food intake and digestibility [52]. Interestingly, the present study observed a positive correlation between ECI and ECD but a negative correlation with AD. This intriguing finding indicates that the elevated conversion and utilization rates might serve as a physiological compensation for the decreased digestibility, which is consistent with previous research [19,53]. These research studies indicated that the nutrients and secondary metabolites between maize varieties stimulate changes in nutritional indexes, as they play a direct role in the growth and development of larvae [39,54].

Nutrient utilization and conversion in plant-feeding insects hinge on digestive enzyme activities in the midgut, reflecting their capacity for digestion, absorption and their consequent impact on growth and development [55,56]. According to differences analysis, different maize varieties can significantly affect enzyme activities. As confirmed by previous studies [57], changes in digestive enzyme activity have been shown to affect the adaptability of *S. frugiperda* to different hosts. In our study, *S. frugiperda* reared on the common maize varieties exhibited higher activities of β-amylase, total-amylase, and lipase. For example, Wuke 113 showed the highest levels of β-amylase and total-amylase activities, whereas Zaocuiwang exhibited the lowest activities for both β-amylase and total-amylase. Pearson correlation analysis showed that β-amylase and total-amylase exhibited significantly negative and positive correlations with fatty acid and moisture, respectively. Similarity exists with previous studies that have shown the amylase activity of *Helicoverpa armigera* Hübner (Lepidoptera: Noctuidae) decreases when reared on plants with high carbohydrate content [58]. Furthermore, higher protein in common maize varieties may lead to higher lipase activity. These observed variations in enzyme levels could be attributed to differences in nutrient composition or secondary metabolite content in the host plants. Our research showed that nutrient utilization efficiency and digestive enzyme activity can interact with each other. Common maize varieties, for example, had higher digestive enzyme activities and RCR but lower ECD and ECI. These results suggest that *S. frugiperda* perhaps needs to enhance digestive enzyme activities in order to digest and absorb food nutrients more effectively, thereby maintaining normal development and aiding in coping with plant defense mechanisms, likely as an adaptive response. Furthermore, we observed that, under conditions of high food intake, *S. frugiperda* gut amylase activity reached its peak [59]. Similar findings were reported on *H. armigera* reared on seven bean cultivars [60]. Previous studies reported that the efficiency of converting digested food into larval biomass relies on the levels of digestive enzyme activities [61]. Therefore, insects possess a mechanism for accurately detecting and quantifying food contents, allowing them to regulate essential digestive enzyme levels [58]. Our previous research found that *S. frugiperda* showed lower fitness in common maize varieties [38]. In general, plants defend themselves against herbivores by producing various digestion inhibitors that limit nutrient uptake. In response to plants’ defense mechanisms, insect herbivores may counter by augmenting digestive enzyme activities or upregulating inhibitor-insensitive enzyme levels in their midgut [62,63]. Moreover, Namin et al. [60] indicated that enzyme-inhibiting components in red kidney beans (*Phaseolus vulgaris* L.) Sayyad could reduce the food conversion rate in *H. armigera* larvae. Consequently, variations in enzyme levels might also arise from differences in insect responses to enzyme inhibitors within host plants. 

Plants typically defend themselves against herbivores by producing toxic secondary metabolites, either constitutively or in response to attack. However, insects have evolved detoxifying enzymes to counteract these defenses [64]. Furthermore, herbivorous insects tend to increase the expression of detoxification enzyme isoforms when feeding on host plants [65,66,67]. In our study, all six tested maize varieties exerted a significant influence on GST, CYP450, and CarE activities in *S. frugiperda*. The activity of detoxification enzymes changes with different host plants, as evidenced by numerous studies. For example, Glutathione S-transferase Slgste1 has been identified as a crucial detoxification enzyme induced by phytochemicals in host plants, potentially playing a role in the host plant adaptation of *Spodoptera litura* Fabricius (Lepidoptera: Noctuidae) [68], which is consistent with the results of this study. Pearson correlation analysis showed that detoxification enzyme activities had a certain internal relationship between nutrient composition and secondary metabolite content. For example, protein, amino acids, tannin, chlorophyll a, chlorophyll b, and carotenoids can significantly influence the activities of detoxification enzymes. Additionally, Gossypol, a compound found in cotton leaves, has been shown to induce several P450 enzymes (CYP9A12, CYP9A14, CYP6B7, CYP6AE14, and CYP321A1) in *H. armigera* [67]. Similarly, in *Manduca sexta* Linnaeus (Lepidoptera: Sphingidae) larvae, increased P450 levels allow for improved detoxification of secondary metabolites [69], leading to enhanced adaptation to host plants [70]. As a result, we hypothesized that differences in GST, CYP450, and CarE activities could be explained by differences in nutrient content and secondary metabolites of different maize varieties and that higher tannin and protein content of maize varieties could lead to increased GST and CYP450 activities. This may also be an adaptation of insects to different host plants during long-term evolutionary development. Insects can actively regulate enzyme activity to maximize the digestion and utilization of the compounds they feed on. Our previous research has shown that the nutrient content and secondary metabolites of different maize varieties can affect the feeding selection, development, and reproduction of *S. frugiperda* [38,39]. In this study, we also found that there are significant differences in nutrient utilization efficiency and the digestive and detoxification enzyme activities of *S. frugiperda* feeding on different maize varieties, and there is a significant correlation between the nutrient content and secondary metabolites of maize. Thus, the nutrient content and secondary metabolites of different corn varieties may be key factors affecting the adaptability of the *S. frugiperda*.

## 4. Materials and Methods

### 4.1. Insect Rearing

*S. frugiperda* larvae were collected from maize crops in Jingyuan County, Gansu, China (36°34′16″ N, 104°40′36″ E) in September 2019. Subsequently, these larvae were exposed to an artificial diet, as described by Wang et al. [71], while the adults were fed with a 10% honey solution. All the insects were reared in a 40 × 40 × 40 cm^3^ insect cage, set under controlled laboratory conditions with a temperature of 25 ± 1 °C, relative humidity of 75 ± 5%, and a photoperiod of 16 h of light and 8 h of darkness. After acclimatization over ten generations, neonate larvae (<6 h old) were isolated into individual 9.0 cm diameter disposable plastic Petri dishes nurtured on a diet of maize seedlings.

### 4.2. Host Plants and Growth Condition

Six maize varieties were selected for early oviposition selectivity based on the high, medium, and low egg-laying rates of *S. frugiperda* to common and special maize varieties [38]. Seeds of common maize varieties (Zhengdan 958, Wuke No. 4, and Wuke 113) and special maize varieties (Zaocuiwang, Baitiannuo, and Ziyunuo) were supplied by Jiuquan Jinhui Agricultural Development Co., Ltd. in Jiuquan, China.

The maize seeds were sterilized separately in 75% (*v*/*v*) ethanol and 1% (*w*/*v*) NaClO for 3 min and 15 min, respectively, then washed five times with sterile water. Subsequently, they were planted on 1/2 Murashige and Skoog (MS) medium containing 1% (*w*/*v*) sucrose and 0.7% (*w*/*v*) agar, with a pH of 5.8 and cultured at 28 °C under a 16-h light/8-h dark regime for the specified number of days until germination. Then, the seeds were sown in pots (15 cm in height by 20 cm in diameter) containing a potting soil mixture (3:1). These pots were subsequently placed in an insect-rearing room maintained at a temperature of (25 ± 1) °C and a photoperiod of 16:8 h (L:D), with a light intensity of 250 μmol m^−2^ s^−1^ and a relative humidity of 65 ± 5%. The *S. frugiperda* larvae were reared on maize plants at the six-leaf stages.

### 4.3. Host Strain Identification and Sequence Alignment

In order to identify the genotype of collected *S. frugiperda* larvae, DNA extraction from a single larva (five larvae for repeated processing) using the E.Z.N.A. Insect DNA Kit (Omega, Norcross, GA, USA). Briefly, samples were homogenized and lysed in a high-salt buffer containing CTAB and extracted with chloroform to remove polysaccharides. Subsequently, a rapid alcohol precipitation step, binding conditions were adjusted, and DNA was further purified using HiBind^®^ DNA Mini Columns. PCR amplification was performed using a 25 µL reaction mix containing 12.5 µL 2× Hieff PCR Master Mix (Yeasen, Shanghai, China), 2 µL of a 10 µM primer mix, 1 µL DNA template, and the remaining volume filled with water. The thermocycling program consisted of an initial denaturation step at 94 °C for 5 min, followed by 31 cycles of denaturation at 94 °C for 30 s, annealing at 55 °C for 30 s, extension at 72 °C for 30 s, and a final extension at 72 °C for 5 min. Amplification of *COI* used the primer pair 101F (5′-TTCGAGCTGAATTAGGGACTC-3′) and 911R (5′-GATGTAAAATATGCTCGTGT-3′), Amplification of the *Tpi* region with the primers 282F (5′-GGTGAAATCTCCCCTGCTATG-3′) and 850R (5′-AATTTTATTACCTGCTGTGG-3′) (synthesized by Tsingke Biotech Co., Ltd., Xi’an, China) [72]. The PCR products were subjected to 1% agarose gel electrophoresis, followed by Sanger sequencing provided by Tsingke Biotechnology Co., Ltd. (Beijing, China). Newly obtained sequences were submitted to the GenBank database. Newly obtained sequences were submitted to the National Center for Biotechnology Information (http://blast.ncbi.nlm.nih.gov/Blast.cgi, accessed on 13 December 2023). It was compared with sequences of other *S. frugiperda* species in GenBank using the nucleotide BLAST program. Multiple alignments were analyzed using the DNAMAN version 6.0.3.99 (Lynnon Corporation, Vaudreuil-Dorion, QC, Canada), published *COI* sequences (The corn biotype HM136586 and rice biotype HM136593) [73], and *Tpi* sequences (The corn biotype KT336237 and rice biotype KT336230) [74] from GenBank were included in our analyses.

### 4.4. Assessment of Nutritional Indices of 3rd Instar Larvae of S. frugiperda on Various Maize Varieties 

Prior to nutritional assessments, third-instar larvae were subjected to a 12-h starvation period to ensure complete defecation, and then the weights of single larvae were recorded. Subsequently, the larvae were provided with sufficient leaves from corresponding maize seedings, with the initial fresh weight of the leaves meticulously documented. After a 48-h feeding period, larvae were isolated, again subjected to a 12-h starvation period, and then weighed to determine their fresh post-feeding mass. Subsequent to feeding, the residual leaves, larvae, and feces were desiccated at 80 °C until a consistent weight was achieved and determined the dry weight. This process allowed for the calculation of the dry weight of both leaves and larvae pre-consumption, adjusted based on the determined drying rate of the host plant material. The experiment was set up with 25 larvae per trial, repeated 5 times. The nutritional indices of *S. frugiperda* larvae were computed following the protocol established by Waldbauer [75].
$$\mathrm{Relative}\;\mathrm{growth}\;\mathrm{rate}\;(\mathrm{RGR})=\frac{G}{B\times T}$$
$$\mathrm{Relative}\;\mathrm{consumption}\;\mathrm{rate}\;(RCR)=\frac{I}{B\times T}$$
$$\mathrm{Efficiency}\;\mathrm{of}\;\mathrm{conversion}\;\mathrm{of}\;\mathrm{ingested}\;\mathrm{food}\;(ECI)=\frac{G}{I}\times 100\%$$
$$\mathrm{Efficiency}\;\mathrm{of}\;\mathrm{conversion}\;\mathrm{of}\;\mathrm{digested}\;\mathrm{food}\;(ECD)=\frac{G}{I-F}\times 100\%$$
$$\mathrm{Approximate}\;\mathrm{digestibility}\;(AD)=\frac{I-F}{I}\times 100\%$$

In the given equation, various parameters are defined as follows: G represents the weight gain of larvae (calculated as G = dry weight of larvae after feeding—dry weight of larvae before feeding), B represents the mean larval body weight during the test period (B = (dry weight of larvae before feeding + dry weight of larvae after feeding)/2), I represents the weight of food eaten during the test period (I = dry weight of leaf before feeding—the dry weight of leaf after feeding), F represents the dry weight of fecal matter during the test period, and T represents the duration of the test period (measured in days). 

### 4.5. Measurement of Digestive Enzyme Activity 

#### 4.5.1. Midgut Dissection and Enzyme Extraction

The sample was prepared with minor modifications based on the methods described by Wang and Qin [76]. Fifth-instar larvae of *S. frugiperda* were collected and subjected to a 6-h period of starvation. Midguts were then dissected on ice and rinsed with chilled physiological saline to remove food remnants. Subsequently, tissues were homogenized in a pre-cooled homogenizer with distilled water. The homogenate was centrifuged at 15,000 rpm for 10 min at 4 °C, and the resultant supernatant was reserved as the enzyme source. This procedure was conducted on groups of 10 larvae, with the entire assay replicated thrice for consistency.

#### 4.5.2. Amylase Activity Assay

The quantification of amylase activity was conducted using the dinitrosalicylic acid (DNS) method [77]. The assay mixture consisted of 20 µL enzyme extract, 100 µL of phosphate buffer (0.02 M, pH 7.1), and 40 µL of 1% soluble starch. The mixture was incubated at 37 °C for 30 min. The enzymatic reaction was then halted by adding 100 µL of DNS reagent, followed by a 10-min boiling step. After cooling, the absorbance was measured at 540 nm to determine amylase activity (BioTek Instruments, Inc., Winooski, VT, USA). One enzyme activity unit is defined as catalyzing the production of 1 mg of reducing sugar per gram of tissue per minute in 30 min at 37 °C.

#### 4.5.3. Lipase Activity Assay

Lipase activity was assessed by a modified version of the method from Bai and Wang [78], using an olive oil emulsion as the substrate. The assay involved pre-mixing 0.5 mL of phosphate buffer (0.025 mol/L, pH 7.5) with 0.4 mL of olive oil, followed by a 10-min incubation at 40 °C. To this, 0.5 mL of the enzyme suspension solution was added, and the incubation continued for another 20 min at the same temperature. The reaction was terminated by the addition of 1.5 mL of 95% ethanol and a drop of 1% phenolphthalein indicator. Followed by titration with 0.05 mol/L NaOH until a reddish color persisted. The volume of NaOH dispensed was recorded, adjusting for the control, which included 95% ethanol added prior to the lipase activity initiation step. The specific activity of lipase is defined as the amount of enzyme required to hydrolyze 1 μmol of fatty acid per minute under pH 7.5 and 40 °C conditions per gram of lipase.

### 4.6. Determination of Detoxification Enzyme Activity 

#### 4.6.1. Enzyme Solution Preparation

Triplicates of five fifth-instar larvae on six maize varieties were homogenized in pre-cooled phosphate buffer (0.1 M, pH = 7.5). The supernatant, extracted post-centrifugation at 12,000 rpm at 4 °C for 10 min, was utilized as the source of enzymes.

#### 4.6.2. Glutathione-S-Transferase (GST) Activity Assay

GST catalyzes the conjugation of 1-chloro-2,4-dinitrobenzene (CDNB) with reduced glutathione (GSH), yielding a dinitrophenyl thioether (GS-DNB) with a high extinction coefficient at 340 nm, facilitating its detection. The GST activity was quantified employing an established protocol [79]. Larvae were homogenized in 1 mL of pre-cooled sodium phosphate buffer (0.1 M, pH 7.2). The homogenate was centrifuged at 10,000 rpm for 30 min at 4 °C, and the resultant supernatant was reserved as the enzyme solution. For the enzymatic reaction, 810 µL of sodium phosphate buffer was mixed with 30 µL of 1-chloro-2,4-dinitrobenzene (CDNB), 50 µL of enzyme solution, and 30 µL of glutathione (GSH). The mixture was then incubated for 5 min at 25 °C. The absorbance was recorded at 340 nm over 5 min with a read 30-s interval using an ultraviolet spectrophotometer (BioTek Instruments, Inc., Winooski, VT, USA). For the control group, an equivalent volume of phosphate buffer is added instead of an enzyme solution. At 25 °C, one enzyme activity unit is defined as the catalysis of 1 μmol of CDNB binding with GSH per gram of sample per minute.

#### 4.6.3. CarE Activity Determination

CarE activity was determined following a previously described method [80]. Briefly, the homogenization of samples occurred in an ice bath using 1 mL of sodium phosphate buffer (0.04 M, pH 7.0), followed by centrifugation at 11,000 rpm for 15 min at 4 °C. The clear supernatant was then prepared for the assay by combining it with 1.8 mL of substrate solution (containing 3 × 10^−4^ M α-NA and 3 × 10^−4^ eserine), 0.45 mL of sodium phosphate buffer (0.04 M, pH 7.0), and 0.05 mL of enzyme solution, incubated for 15 min at 30 °C. The reaction was terminated with 0.9 mL of dye reagent (1% of Fast blue B salt: 5% of sodium dodecyl sulfate solution = 2:5). The optical density (OD) for the reaction mixtures was measured at 600 nm (BioTek Instruments, Inc., Winooski, VT, USA). One enzyme activity unit (U) is defined as the increase of 1 absorbance unit per mL reaction system per gram of tissue per minute, and enzyme activity was expressed as U/g. 

#### 4.6.4. CYP450 Activity Determination

The procedure for CYP450 activity measurement was adapted from a previously reported method [81]. The reaction mixture consisted of 705 μL of Tris-HCl buffer, 25 μL of 2 mmol/L of 7-ethoxycoumarin (7-EC), 20 μL of 6 mmol/L NADPH, and 250 μL of enzymatic suspension, incubated for 30 min at 30 °C. The reaction was halted using 300 μL of trichloroacetic acid and subsequently centrifugation at 12,000 rpm for 3 min. The fluorescence of the resultant 7-hydroxycourmarin was measured with excitation at 368 nm and emission at 456 nm (BioTek Instruments, Inc., Winooski, VT, USA). As the control, the same amount of enzyme was added after introducing trichloroacetic acid.

### 4.7. Statistical Analysis

Data from these experiments were processed and analyzed using IBM SPSS Statistics version 23.0 (Chicago, IL, USA). The results are expressed as means ± standard error (SE). The effects of various maize cultivars on the nutritional indices, as well as digestive and detoxification enzyme activities, were assessed through a one-way analysis of variance (ANOVA). Post-hoc comparisons were made using Duncan’s multiple-range test to identify specific differences between groups. A correlation analysis was performed to explore the relationship between the chemical substances of maize leaves, as measured in our previous study [39] (Appendix A), and the nutritional and enzymatic parameters observed in the current study. These analyses employed the Pearson correlation coefficient to quantify the degree of association, with an accompanying *p*-value indicating its statistical significance. Thresholds for significance were set at *p* ≤ 0.05 for significant differences and *p* ≤ 0.01 for extremely significant differences. The figures related to nutritional indices and enzyme activities were generated using GraphPad Prism 6 (GraphPad Software, Boston, MA, USA).

## 5. Conclusions

This comprehensive study provided valuable insights into the complex interactions between *S. frugiperda* and various maize varieties. Our analysis revealed that both the *COI* and *Tpi* genes identified *S. frugiperda* as the corn strain. Our findings also reveal the significant impact of host plant diversity on the nutritional indices and digestive and detoxification enzyme activities in *S. frugiperda* larvae. The research revealed that special maize varieties facilitate higher nutrient assimilation efficiency in *S. frugiperda* compared with common maize varieties. This indicates a potential adaptation strategy of the pest to different host plant characteristics. The elevated activities of amylase and lipase enzymes in larvae reared on common maize varieties suggest a nuanced physiological adaptation to optimize nutrient extraction from these host plants. The correlation analysis further established a strong link between the maize plants’ nutritional content and secondary metabolites and the physiological responses of *S. frugiperda*, highlighting the pest’s ability to adjust its metabolic processes in response to the chemical composition of its food source. These findings contribute significantly to the understanding of *S. frugiperda*’s adaptability and resilience against different maize cultivars. They also offer a crucial theoretical basis for developing sustainable pest management strategies. By considering the diversity of maize crops and understanding the adaptive mechanisms of *S. frugiperda*, more effective and environmentally sustainable approaches to pest control can be devised. This study, therefore, not only enhances our knowledge of insect-plant interactions but also provides practical insights for agronomists and farmers in managing *S. frugiperda* infestations more effectively. 

## Figures and Tables

**Figure 1 plants-13-00597-f001:**
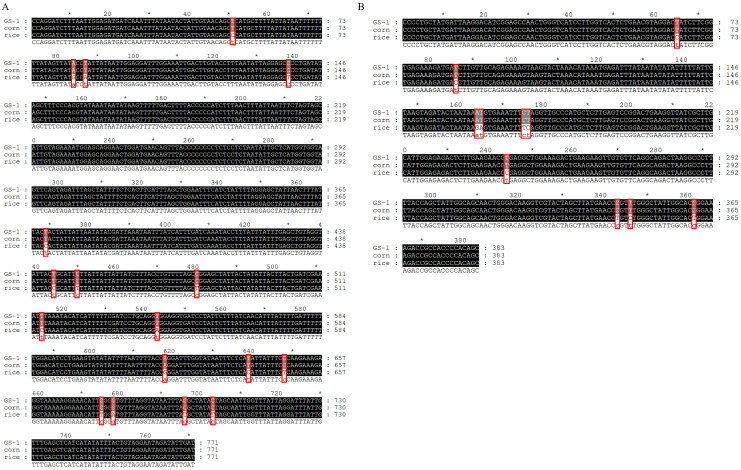
Multiple alignments of *COI* and *Tpi* gene fragments in *S. frugiperda* individuals. (**A**) *COI* gene fragments; (**B**) *Tpi* gene fragments. GS-1 represents the first sequence of collected *S. frugiperda* in Gansu. Corn and rice represent the corn and rice strain. In the last line, capital letters represent identical sequences, while lowercase letters represent different sequences. The asterisk (*) indicates a difference of 10 base pairs compared to the preceding sequence, and the red box represents variant bases.

**Figure 2 plants-13-00597-f002:**
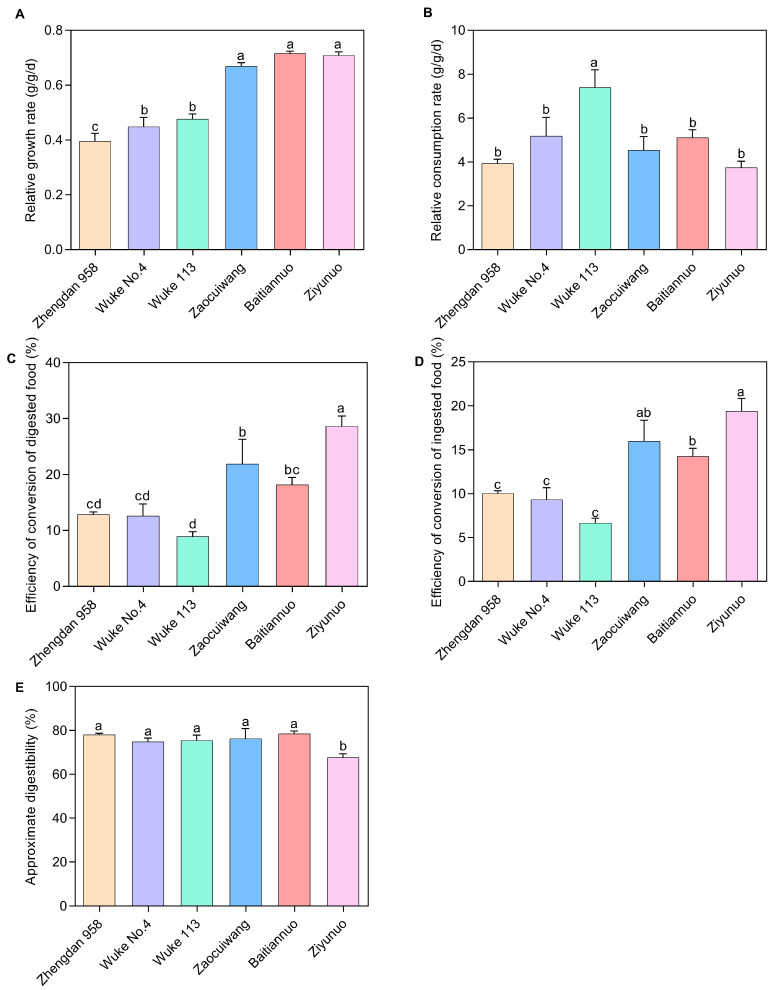
Effects of six maize varieties on nutritional indices of 3rd-instar larvae of *S. frugiperda*. (**A**) Relative growth rate; (**B**) Relative consumption rate; (**C**) Efficiency of conversion of digested food; (**D**) Efficiency of conversion of ingested food; (**E**) Approximate digestibility. Data presented are the mean ± standard error (SE) of five replicates, *n* = 25. Distinct lowercase letters within a histogram indicate significant differences among maize varieties (Duncan’s test, *p* < 0.05). The X-axis represents various maize varieties. The common maize varieties include Zhengdan 958, Wuke No. 4, and Wuke 113, while the special maize varieties consist of Zaocuiwang, Baitiannuo, and Ziyunuo.

**Figure 3 plants-13-00597-f003:**
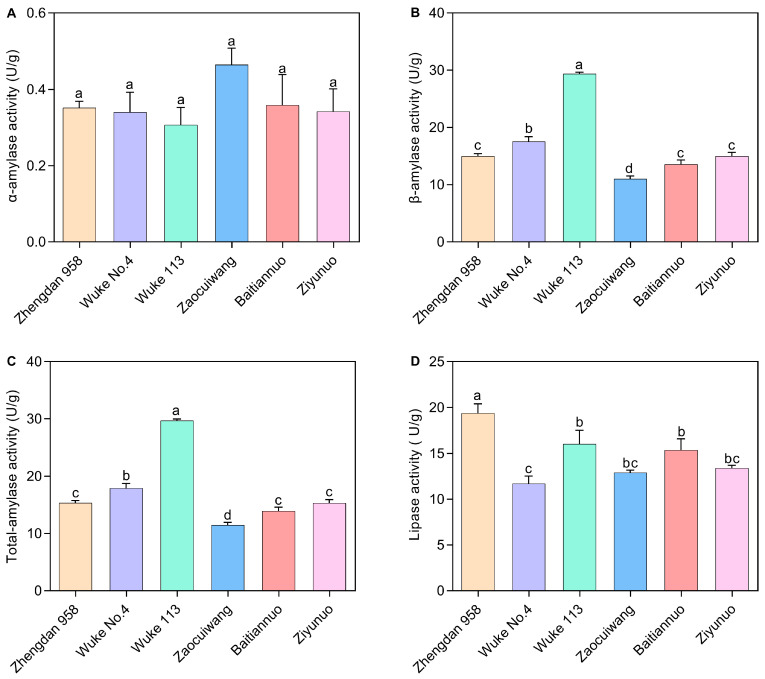
Digestive enzyme activity of *S. frugiperda* larvae on six maize varieties. (**A**) α-amylase activity; (**B**) β-amylase activity; (**C**) Total-amylase activity; (**D**) Lipase activity. The presented data are expressed as the mean ± SE of three replicates, *n* = 10. Significant distinct values at the 0.05 level are marked by different lowercase letters. The X-axis represents various maize varieties. The common maize varieties include Zhengdan 958, Wuke No. 4, and Wuke 113, while the special maize varieties consist of Zaocuiwang, Baitiannuo, and Ziyunuo.

**Figure 4 plants-13-00597-f004:**
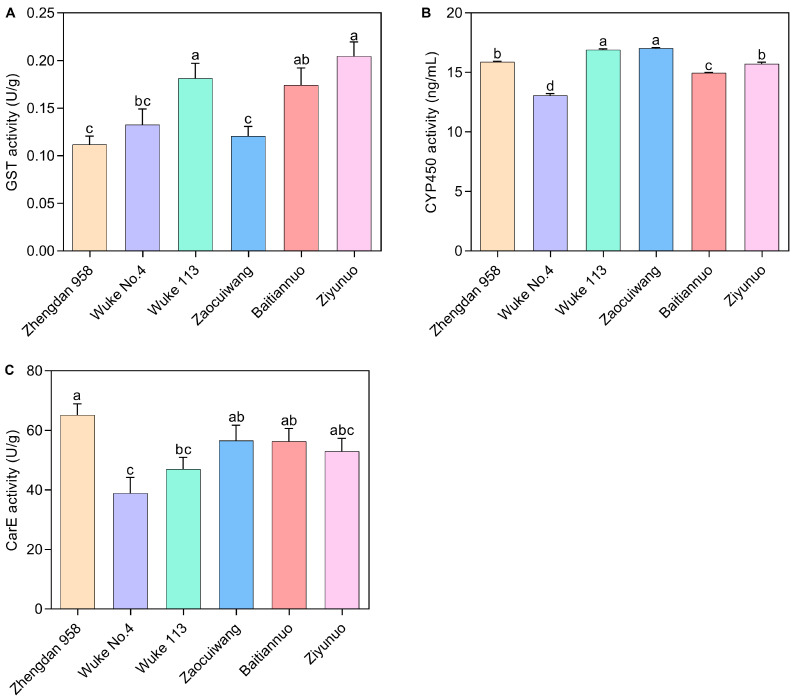
Effects of six maize varieties on detoxification enzyme activity of *S. frugiperda* larvae. (**A**) GST activity; (**B**) CYP450 activity; (**C**) CarE activity. The data represent the mean ± SE of three replicates, *n* = 5. Lowercase letters indicate a significant difference among the six maize varieties (Duncan’s test, *p* < 0.05). The X-axis represents various maize varieties. The common maize varieties include Zhengdan 958, Wuke No. 4, and Wuke 113, while the special maize varieties consist of Zaocuiwang, Baitiannuo, and Ziyunuo.

**Figure 5 plants-13-00597-f005:**
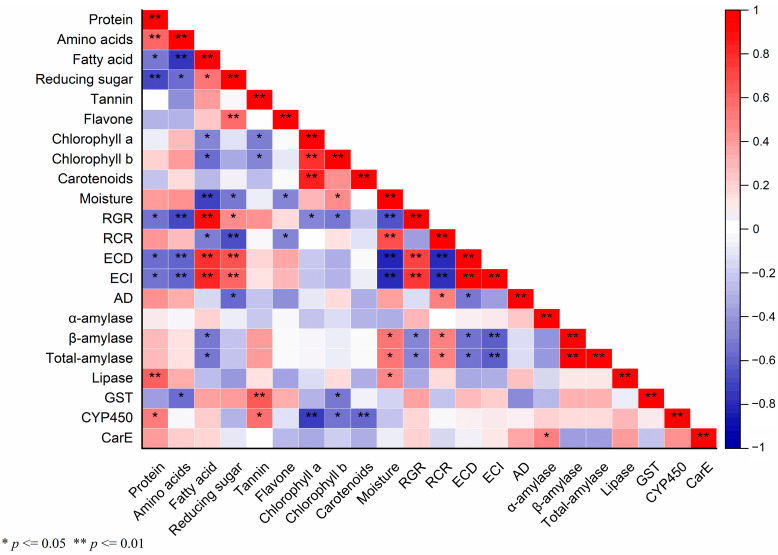
The correlation matrix of different indices is based on the Pearson correlation coefficient. Colors from blue to red represent a correlation from extremely negative to positive. Significance or extreme significance are indicated by *p* ≤ 0.05 or *p* ≤ 0.01, respectively.

## Data Availability

The datasets in this study are available from the corresponding author upon reasonable request.

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
