# Peer review of "Molecular Characterization Analysis and Adaptive Responses of *Spodoptera frugiperda* (Lepidoptera: Noctuidae) to Nutritional and Enzymatic Variabilities in Various Maize Cultivars"

_plants, 2024, doi:10.3390/plants13050597_

Round 1

Reviewer 1 Report

Comments and Suggestions for Authors

Comment for authors.

In their analyses, researchers provide new, interesting data on the phenology, physiology, and biochemistry of the invasive pest Spodoptera frugiperda in China concerning corn varieties grown in China. Thus, they meet the Chinese agricultural economy's local needs and understand its biology in Chinese conditions. However, for this reason, the publication is mainly hermetic for readers from Europe or America, among other things, due to data published in Chinese-language journals unavailable in global databases.

The planned experiment, part and continuation of a more extensive study, is carried out correctly and does not raise any major concerns. The methods used are correct, as are the statistical analysis tools used. The manuscript requires only minor corrections to the figures, mainly for better understanding. It is also necessary to make a few additions and define some concepts to make the article clear in its content and readable to all its recipients worldwide. Detailed notes and comments are provided below.

The article can be submitted for publication after making the necessary corrections and supplementing the data from Chinese journals.

Comments.

·         Six varieties of corn were selected for the experiment. There is no description of them. All varieties appear to be characteristic of Chinese agriculture. The authors characterize three of them as "common maize varieties" and the remaining three as "special maize varieties". These concepts need to be defined. This division is not considered in any way in the figures - each variety is assigned a meaningless bar color. If this distinction is important, it should be reflected in the presentation of the results.

·         Why was the 3rd stage of S. frugiperda selected for study? Perhaps the results for the last developmental stage, when the caterpillars are in the hyperphagic stage, would be more reliable. This needs to be justified.

·         The charts lack a description of the "X" axis - corn varieties.

·         Figure 2 - does "five replicates" mean the number of trials to be analyzed - dependent variables (n = 5?).

·         Figure 3 - activity unit "U" should be defined - this can be provided in the marking methods in the M&M chapter.

·         Figures 3 and 4: The caption does not contain information about the number of dependent variables (n) that are the basis for the statistical analysis.

·         Pearson's correlation: lines 238-240. There is no information about leaf parameters. These parameters should be characterized, and their values should be provided for individual corn varieties considered for correlation calculations. Were the correlations made for each corn variety separately or collectively for all varieties? In lines 553-536, the authors refer to the publication by Zhang et al., 2021. However, the bibliographic note in "References" is incorrect. The publication was published not in "Plant Protection" but in "China Plant Protection" and in Chinese. There is a "Plant Protection" tab on the website of this publishing house. However, the article cannot be found here under the indicated number 41 and the year 2021. There are six numbers from this year, volume 47. This article contains key information about the data used for the correlation analysis as well as (it should be concluded) information about corn varieties. Without this data, the reader must take the correlation analysis "on faith", and the entire publication is understandable only to a Chinese reader. The publication should clearly indicate another, fully available source of this information or include the necessary data from this or other Chinese publications in the article submitted for publication (perhaps as an annex). Without meeting this condition, I do not recommend submitting the publication for publication. Another publication may contain this information - Lu et al., 2020 (Journal of Environmental Entomology) - item 55 "References", is also a Chinese journal.

Abstract: Line 18 – the term “original” is ambiguous – what do the authors mean?

Line 84 (and others) – in the name of the glutathione-S-transferase enzyme – S should be written in italics.

Line 177 – either a comma or a capitalized "while".

Line 241 and the following: insert a space after the = sign.

Lines 459 and 462 – use a subtraction sign, not a hyphen.

Reviewer 2 Report

Comments and Suggestions for Authors

Reviewer's opinion about MS, entitled "Molecular Characterization Analysis and Adaptive Responses of Spodoptera frugiperda (Lepidoptera: Noctuidae ) to Nutritional and Enzymatic Variabilities in Various Maize Cultivars.

General comments: The study is fascinating and novel. The authors are employed in the effect of different maize cultivars on the specific biotype of Spodoptera frugiperda with the help of a wide range of analyses from plant composition analyses to insect physiology.  However, there are some major concerns.

Specific remarks:

-             Firstly, the name of the target organism should be used uniformly: English or scientific. I suggest to prefer the abbreviated scientific name.

-             The titles and subtitles should be rewritten with non-capital letters; please replace them.

-             Some paragraphs should be moved to the proper chapter of this study because it is placed in the wrong sections, such as:

lines 100-105 moves to the MM

lines 106-110 moves to the Results

lines 121-123 moves to the MM

lines 284-293 moves to the Introduction

-             Several captions are missing in Figure 1, such as: What means „GS1”, What is represented in the last line of the nucleotide enumerations? The main problem is the missing explanation of the listed sequences in MM. Which database were used in this examination? The scientific uniqueness is questionable, as only the maize biotype is found in China, and several publications reported

Besides, it would be interesting to analyse corn biotypes on secondary host plants.

-             The nutrient content analysis of different maize types is also missing in Results and MM. So, an additional section and figure should be placed into Results and MM.

-             In chapter 4.2, Host Plants, a precise plant breeding protocol should be supplemented, such as wavelengths, light intensity, humidity. date of sowing, sowing parameters, parameters of the plant growth chamber.

-             Chapter 4.3, Host Strain Identification, does not describe exactly how the sequencing was done, the precise protocol is needed.

-             Chapter 4.5 does not name the instrument used to measure the absorbance! These data should be added to the whole manuscript precisely. 

-             In chapter 4.7, lines from 526 to 528 it mentions the genes used but there is no GENEID next to it, please replace them and add lines from 122-124.

-             The exact enzyme activity measurement protocols should be added.

-             Definition of the displayed nomenclature is inappropriate: mature marker genes. This should be corrected to strain markers.

-             The content of acknowledgement is unusual. The thanksgiving to the editor and the reviewers is unnecessary. Please delete it, and rephrase the section.

-             The names of the cited journals in the reference are very varied, please uniformly use the journal names based on the requirements of Plants MDPI.

In summary, the displaying work is valuable and informative. It can be useful for the theoretical plant production experts. But in the present form of the manuscript implies several mistakes which should be corrected in any case. Therefore, after the major revision the manuscript can be published in Plants MDPI. 

Round 2

Reviewer 2 Report

Comments and Suggestions for Authors

Dear Authors,

Critical comments have been taken into account and the authors have endeavoured to respond to them as far as possible. I appreciate the efforts put into improvement. Overall, the content of the manuscript shows a great deal of work, which is reflected in the quality of the current version.The manuscript is now acceptable in this form and I recommend it for publication.

I congratulate on your achievement.

Author Response

Dear editor and reviewers:

Thanks very much for your kind work and consideration on publication of our paper. On behalf of my co-authors, we would like to express our great appreciation to editor and reviewers.

Thank you and best regards.

Yours sincerely,

Qiang-Yan Zhang

E-mail: zhangqiangyan2@163.com